# Data sharing upon request and statistical consistency errors in psychology: A replication of Wicherts, Bakker and Molenaar (2011)

Aline Claesen[1]*, Wolf Vanpaemel[1]*, Anne-Sofie Maerten[1], Thomas Verliefde[2], Francis Tuerlinckx[1], Tom Heyman[1,3]

1 Faculty of Psychology and Educational Sciences, KU Leuven, Leuven, Belgium, 2 Department of Psychology, University of Tuebingen, Tuebingen, Germany, 3 Faculty of Social Sciences, Leiden University, Leiden, The Netherlands

* aline.claesen@kuleuven.be (AC); wolf.vanpaemel@kuleuven.be (WV)

**Data Availability Statement:** The secondary data from Wicherts et al. (2011) and Vanpaemel et al. (2015) cannot be shared because of ethical

## Abstract

Sharing research data allows the scientific community to verify and build upon published work. However, data sharing is not common practice yet. The reasons for not sharing data are myriad: Some are practical, others are more fear-related. One particular fear is that a reanalysis may expose errors. For this explanation, it would be interesting to know whether authors that do not share data genuinely made more errors than authors who do share data. (Wicherts, Bakker and Molenaar 2011) examined errors that can be discovered based on the published manuscript only, because it is impossible to reanalyze unavailable data. They found a higher prevalence of such errors in papers for which the data were not shared. However, (Nuijten et al. 2017) did not find support for this finding in three large studies. To shed more light on this relation, we conducted a replication of the study by (Wicherts et al. 2011). Our study consisted of two parts. In the first part, we reproduced the analyses from (Wicherts et al. 2011) to verify the results, and we carried out several alternative analytical approaches to evaluate the robustness of the results against other analytical decisions. In the second part, we used a unique and larger data set that originated from (Vanpaemel et al. 2015) on data sharing upon request for reanalysis, to replicate the findings in (Wicherts et al. 2011). We applied *statcheck* for the detection of consistency errors in all included papers and manually corrected false positives. Finally, we again assessed the robustness of the replication results against other analytical decisions. Everything taken together, we found no robust empirical evidence for the claim that not sharing research data for reanalysis is associated with consistency errors.

## Introduction

A major benefit of sharing research data is that it allows the scientific community to verify the empirical results and scientific findings, and to further build upon the published work. Moreover, as Bosma and Granger [1] showed, there are also ethical implications of data sharing.

restrictions. Due to the nature of these studies, it was not possible to ask contacted authors for informed consent, while it concerns sensitive information. The study by Wicherts et al. (2011) was approved by the Ethics Committee of the Psychology Department of the University of Amsterdam. For data request we refer to the authors, who indicated that data will be shared upon request (we contacted Jelte Wicherts) at the email (J.M.Wicherts@tilburguniversity.edu). The study by Vanpaemel et al. (2015) was approved by the Faculty of Psychology and Educational Sciences of the University of Leuven (i.e., by SMEC, the social and societal ethics committee) under the restriction that they would not disclose who shared their data and who did not, for "Making the response to our data request public would constitute a breach of confidentiality." (Vanpaemel et al., 2015) - The primary data (i.e., documented errors in papers) are available from the Open Science Framework on our project page: https://osf.io/k9nej/?view_only=cd1e9db044814b77b1964398b20b18c9 - The secondary data from Nuijten et al. (2016) is available from the Open Science Framework: https://osf.io/axfbn/.

**Funding:** A.C. was funded by the Special Research Fund of KU Leuven under Grant C14/19/054 (https://researchportal.be/en/project/derailment-affective-system-measurement-early-detection-and-underlying-mechanisms). A.M. was funded by the Fonds Wetenschappelijk Onderzoek (FWO, https://www.fwo.be/en/) under Grant 11C9522N. T. V. was funded by the Deutsche Forschungsgemeinschaft (DFG, German Research Foundation, https://www.dfg.de/en/) under Grant GRK 2277 "Statistical Modeling in Psychology". The funder had no role in study design, data collection and analysis, decision to publish, or preparation of the manuscript.

**Competing interests:** The authors have declared that no competing interests exist.

Despite all these benefits, Wicherts et al. [2] found that researchers are not keen on sharing their data when they requested data for reanalysis to investigate the influence of outliers, as they received data from only 27% of the contacted authors. Almost a decade later, after the onset of the replication crisis, Vanpaemel et al. [3] found that data sharing was still a precarious practice in psychology when they requested data for a Bayesian reanalysis and received data from 38% of the contacted authors. Researchers invoke a myriad of reasons for not sharing data [4, 5]. Houtkoop et al. [4] revealed several barriers to data sharing. Some are practical, like ethical restrictions or time constraints. Other barriers are related to fear, such as the fear that the data will be misinterpreted or scooped. One particular fear is that a reanalysis may expose errors, which potentially alter the conclusions.

In line with this last explanation, Wicherts, Bakker and Molenaar [6] hypothesized that authors that do not share data made more errors than authors who do share data. Testing this hypothesis is not straightforward, since it is not possible to reanalyze unavailable data. Wicherts et al. [6] circumvented this issue by examining errors that can be discovered based on the published paper only, without access to the underlying data. More specifically, they searched papers for triplets of the test statistic, the degrees of freedom, and the $p$-value. Afterward, they recalculated the $p$-value based on the reported test statistic and degrees of freedom (and, if needed, additional information, such as whether the test was one- or two-tailed). Reporting errors were detected by comparing the recalculated $p$-value with the reported $p$-value.

In the current paper, we use the following terminology. If the recalculated $p$-value differs from the reported $p$-value (taking differences due to rounding into account), then this is labeled a consistency error. In case the recalculated $p$-value leads to a different decision regarding the rejection of the null hypothesis than the reported $p$-value (assuming $\alpha = .05$), this is termed a decision error. In this approach, decision errors are a subset of consistency errors. Definitions and examples of consistency errors are included in Table 1. Consistency errors may be the result of questionable research practices, for example when the $p$-value is incorrectly rounded down to reach statistical significance (e.g., see [7]; this would be a decision error). Nevertheless, inconsistencies do not necessarily indicate such practices. They can also result from mere typos in the test statistic, degrees of freedom, or $p$-value, which seem to happen commonly [8]. Vice versa, underlying errors do not necessarily manifest themselves through consistency errors. The absence of a consistency error is no guarantee of the absence of underlying errors.

For results reported with a $p$-value below .05, Wicherts, et al. [6] found a higher prevalence of consistency errors in papers for which the research data were not shared. They also found a relationship between the presence of decision errors and sharing research data upon request. Data were shared for none of the papers with decision errors. In contrast, Nuijten et al. [9] found no convincing evidence for such a relationship in three large studies. They hypothesized

**Table 1. Definitions and examples.**

| Term | Synonym | Definition | Example |
|---|---|---|---|
| consistency error | reporting error; inconsistency | $p$ and $p'$ are inconsistent | $F(1, 78) = 9.10, p < .001$; $p' = .0035$ |
| decision error | reporting error concerning $p < .05$; gross inconsistency | $p$ and $p'$ are inconsistent, and lead to a different decision (i.e., accept/reject $H_0$ assuming $\alpha = .05$) | $F(1, 88) = 3.07, p = .02$; $p' = .0832$ |

*Note*:

$p'$ denotes recalculated $p$-value.

that both journal policies that encourage data sharing and the actual availability of data would be associated with fewer consistency errors. In their first study, they compared the prevalence of consistency errors in papers from two similar journals, of which one included the recommendation to publish open data, whereas the other one did not. They also compared papers for which data was actually available with those for which no data was available. They found no evidence for an effect of open policy nor an effect of open data on consistency errors. In their second study, they compared papers from a journal that required open data and a journal that recommended open data, and they found an effect of journal policy on consistency errors, but not on decision errors. Again, they compared papers for which data was available with papers for which this was not the case, and they found no effect of open data on consistency errors. Finally, they compared papers with and without an Open data badge from the same journal. Again, they found no effect of journal policy (before and after implementation of Open practice badges) and, upon the comparison of papers that had data available with those that did not, no effect of open data on consistency errors. Overall, they did not find convincing evidence for an effect of the journal policy regarding data sharing, nor for a relation between the availability of the data and consistency errors.

This discrepancy in results between the studies of Wicherts et al. [6] and Nuijten et al. [9] could be explained by methodological differences. A first difference is that Nuijten et al. [9] operationalized data sharing by considering different data sharing policies (i.e., the recommendation versus the requirement to publish with open data) and by the public availability of data, while Wicherts et al. [6] focused on data sharing upon request for a reanalysis. Authors might have the impression that it is more likely that existing errors are exposed when data are requested for reanalysis than when the data are openly available (certainly when the open data are not reusable, see [10]). Another important difference is that, in the study of Wicherts et al. [6], statistics were extracted and evaluated manually, whereas, in the studies by Nuijten et al. [9], an automated procedure (i.e., the R package *statcheck* [11]) was employed. This procedure allowed them to increase the sample size drastically. The procedure avoids human errors, but may fail to detect all statistics and cannot take context into account, like a human would do, in evaluating consistency errors [12].

That being said, given the ambiguity about the relation between errors and data availability, and the fact that the study by Wicherts et al. [6] is often referred to without a reference to the studies by Nuijten et al. [9] (e.g., [13]), there is a need for a replication study. Collecting data like Wicherts et al. [2] did, is a very time-consuming endeavor, which requires contacting a large number of authors with a request for data and might require a lot of time for some of the contacted authors as well. Fortunately, Vanpaemel et al. [3], who requested data from papers published in the 2012 volume of four APA journals, already have collected a data set that is perfectly suitable as a replication data set. In this paper, we will use these data for a replication attempt of the findings by Wicherts et al. [6]. More specifically, we investigated the relationship between sharing research data upon request for reanalysis on the one hand, and consistency errors in the reported statistical results on the other hand. The remainder of this paper is organized as follows. In the first part, we focus on the original study by Wicherts et al. [6]. We start by investigating the reproducibility of their results: Do we find the same statistical results if we perform their analyses again on their data? Because there are often numerous viable ways to conduct a data analysis (i.e., researcher degrees of freedom [14]), any data analysis which reports a single pathway provides an incomplete picture. We continue with an investigation of the robustness of their results against other reasonable analytical decisions, by carrying out several alternative analytical approaches (i.e., a multiverse analysis; see Steegen et al. [15]). In the second part of this paper, we describe our replication attempt, in which we studied the same hypotheses as Wicherts et al. [6], but with the data collected by Vanpaemel et al. [3].

Throughout the paper, we refer to the data, analysis, and results, as they are reported by Wicherts et al. [6] with *original data*, *original analysis*, and *original results*, respectively. We refer to our reproduction of the original analysis and results as the *reproduction analysis* and *reproduction results*. Finally, we refer to the selection of our data according to the inclusion criteria from the original study as the *replication data*. The *replication analysis* is the original analysis applied to the replication data, and the *replication results* are the results from the replication analysis.

## Disclosure

All analyses are conducted in R in R version 4.1.2 [16] except when specified otherwise. All R scripts can be found on the Open Science Framework (OSF) in this OSF project: https://osf.io/k9nej/?view_only=cd1e9db044814b77b1964398b20b18c9 Due to ethical considerations, not all data for this paper can be shared. In the first part of this paper, we use the data from the study by Wicherts et al. [6]. This study had been approved by the Ethics Committee of the Psychology Department of the University of Amsterdam, under the restriction that they cannot make public who did or did not share data with Wicherts et al. [2], because they could not ask the contacted authors for informed consent due to the purpose of the study. The documented errors are reported in Wicherts et al. [6]. In the second part of this paper, we use the data from the study by Vanpaemel et al. [3]. This study had been approved by the Ethics Committee of the Faculty of Psychology and Educational Sciences of the University of Leuven, under the restriction that it would not be disclosed who shared their data and who did not, because, again, the contacted authors could not be asked for informed consent due to the purpose of the study. Similar to Wicherts et al. [6], we can share the documented errors: the *statcheck* output and the summary of the manually corrected statcheck output can be found here: https://osf.io/6zr5q/?view_only=3b338aa87cf04b0c998ea989c8b79122. In the discussion, we apply data from Nuijten et al. [12], which can be found here: https://osf.io/axfbn/. Note that for our analysis, this data needs to be combined with the data from Wicherts et al. [6]. All results reported in the main text and discussion are independently co-piloted by TV [17]. All results are reproduced based on the raw data and the description of the methods provided in this paper. The co-pilot's analyses can be found here: https://osf.io/ajp92/?view_only=ca171229f9ae4320bab5d4ad689c7e80. This study has been approved by the social and societal ethics committee at KU Leuven. The reference number of our application is G-2021–4507.

## The reproduction of the study by Wicherts and colleagues [6]

Wicherts et al. [2] requested raw research data from 141 empirical articles published in the last two 2004 issues of four APA journals: *Journal of Personality and Social Psychology* (*JPSP*), *Developmental Psychology*, *Journal of Consulting and Clinical Psychology*, and *Journal of Experimental Psychology: Learning, Memory, and Cognition*(*JEP:LMC*). The authors were asked to share their data in order to assess the robustness of the results to outliers. Although all contacted authors had signed the APA Certification of Compliance With APA Ethical Principles and thus have agreed to provide their data to other researchers for reanalysis (given that there are no ethical or legal restrictions), 73% did not share their data. In a subsequent study, Wicherts et al. [6] found a relationship between data sharing and both consistency and decision errors in a subset of these papers: Errors were less prevalent in papers for which the data were shared.

## Materials and methods

**Sample and variables.**   Instead of using the data from all 141 papers, Wicherts et al. [6] restricted their analysis to 49 papers from two APA journals (*JPSP* and *JEP:LMC*). They reported three reasons for using this smaller sample. First, in this subsample, the data-sharing rate was higher. Second, no corresponding authors declined due to propriety rights, ethical reasons, or because they were part of an ongoing project. Finally, the studies in these journals were more homogeneous regarding design and analysis (i.e., lab experiments). Focusing on the 49 selected papers, for 21 (43%) of these papers, data were shared. In the next step, Wicherts et al. manually extracted per paper unique triplets of test statistics, degrees of freedom and *p*-values (resulting from the application of NHST), and extra information such as whether the test was one- or two-tailed. They only included *t*, *F*, and $\chi^2$ test statistics reported in the results section with a *p*-value below .05. This selection resulted in 1148 triplets. In order to evaluate the consistency within triplets, all *p*-values were recalculated based upon the reported test statistic and degrees of freedom (and accessible S1 File) and compared with the reported *p*-values (using Microsoft Excel 2008 for Mac, version 12.1.0 [18]). Wicherts et al. [6] found 49 (4%) consistency errors, of which 10 (1%) were decision errors. The raw data from Wicherts et al. [6] are not publicly available, but we requested and received the data from Jelte Wicherts for our reanalysis and replication attempt.

**Statistical analysis.**   To start, we attempted to reproduce the results of the two crucial analyses (out of the four reported analyses in total) from Wicherts et al. [6]. In a first analysis, Wicherts et al. investigated the relation between the prevalence of consistency errors and data sharing. A negative binomial regression with a log link was applied to model the number of consistency errors per paper, the outcome variable, as a function of the following three predictors: (1) whether the research data were *shared* (this is the target predictor, coded with 1 for shared and 0 for not shared), (2) the natural logarithm of the number of extracted statistics per paper, and (3) the square root of the mean of recalculated *p*-values that were below .05 per paper. No interactions were included. Because there can be more consistency errors in papers with more reported statistics, the role of the second predictor is to provide a normalization. This predictor could also have been included as an exposure variable, in which case no coefficient has to be estimated, but we follow Wicherts et al. [6] who decided to include it as a separate predictor (and thus estimated the regression coefficient). The hypothesis test was two-tailed, with $\alpha$ = .05. The original analysis was conducted in SPSS 18.0 [19] and the authors reported that the *Zelig* package in R (no version specified) returned similar results. In both cases, a robust variance estimator to find the standard errors was used. Because of the log link, the natural logarithm of the mean is a linear function of the predictors. Wicherts et al. [6] estimated that if the data were shared, the expected log count of consistency errors decreased by 0.83, everything else held constant, which implies a decrease of 56% in consistency errors ($e^{0.83}$ = .44). We reran this analysis on the original data in R with *MASS* (7.3.54) [20], and we employed *sandwich* (version 3.0.1) [21] for the robust variance estimator (i.e., a heteroscedasticity-consistent estimator, type HC0, see Section 3 in S1 File), because the *Zelig* package is obsolete.

In a second analysis, Wicherts et al. [6] studied the relationship between data sharing and the presence of decision errors. Because both variables of interest are binary, a two-by-two table was constructed and the association was tested using Fisher's exact test, which is a suitable approach for small samples (i.e., there were only seven papers with at least one decision error). The hypothesis test was two-tailed and the chosen level of significance for the hypothesis test was $\alpha$ = .05. Using these criteria, Wicherts found a statistically significant relation

between data sharing and making at least one decision error (i.e., $p$ = .015). We also repeated this analysis on the original data in R.

The reproducibility of the results was evaluated by the categorization suggested by Artner et al. [22], which has two dimensions: correctness and vagueness. The results are either correct, meaning that they can be reproduced without conflicting with the reported underlying calculations, or incorrect. As recommended by Artner et al. [22], we allowed for a margin of error due to software differences and rounding. We evaluated vagueness as either high or low, depending on the ambiguity of the reported information about the underlying calculations. Information is ambiguous if there are multiple possible interpretations of how to perform the calculations.

Besides reproducibility, we assessed the robustness of the reproduction results to some alternative analytical choices through a multiverse analysis [15]. In particular, we varied five binary decisions in the original analysis by Wicherts et al. [6], creating 32 specifications to test the first hypothesis. The first four decisions concerned (1) whether to estimate the variance with a robust or standard variance estimator (i.e., the standard error provided by the *MASS* package), (2) whether to include the natural logarithm of the number of statistics per paper as a predictor (with a coefficient to be estimated) or as an exposure variable (without coefficient), (3) whether or not to include the square root of the mean of recalculated $p$-values per paper as a predictor, and (4) whether or not to include the journal the paper was published in as a predictor (this predictor was not included in the original study). These four decisions lead to 16 specifications. A final decision specified whether or not to exclude retracted articles from the analyses (as was suggested to us by Jelte Wicherts in personal communication, August 13, 2020; see Section 1 in S1 File for more details). In a former analysis, we also evaluated other count models besides the negative binomial regression. We do not include those here, because of computational and model fit issues. For transparency, we included the preliminary results of these initial analyses (which were not co-piloted) in Section 4 in S1 File.

For the second hypothesis, we varied three decisions. The first one was the choice of statistical test: the Fisher's exact test, the chi-squared test, or the G-test. Secondly, we tested both the relationship between data sharing and the presence of decision errors per paper, and between data sharing and the number of decision errors per paper. Finally, we either included or excluded retracted articles from the analyses.

## Results

**Reproduction analysis.** Jelte Wicherts shared the original data with SPSS syntax and an R script with corresponding output for all analyses. In SPSS [23], we exactly reproduced the original results by using the original SPSS syntax. In R, we exactly reproduced the point estimates of the negative binomial regression coefficients. The estimates for the standard errors slightly differed from the original results. A possible explanation is that we used a different robust variance estimator than the one used in SPSS. Our reproduction results in R also slightly differed from the original results in R, because we opted for another robust variance estimator (a Heteroscedasticity-Consistent Covariance Matrix Estimation (vcovHC) instead of a Heteroscedasticity and Autocorrelation Consistent (HAC) Covariance Matrix Estimation, because the latter is more suitable for ordered data, such as time series). More information about the differences in standard error estimates is provided in Section 3 in S1 File. Given that the differences between the original and our reproduction results are negligible, we conclude that the original results are reproducible (see Table 2, to be compared to Table 2 in Wicherts et al. [6], also reported in Section 1 in S1 File). The original conclusion remains: If the data were shared, the expected log count of consistency errors decreased by .83, everything else held constant. This

**Table 2. Reproduction of the relationship between data sharing and consistency errors.**

|  | $\hat{\beta}$ | *SE* | *z* | *p* | **2.5%** | **97.5%** |
|---|---|---|---|---|---|---|
| intercept | -2.76 | 1.19 | -2.32 | 0.02 | -5.10 | -0.43 |
| shared | -0.83 | 0.37 | -2.27 | 0.02 | -1.55 | -0.11 |
| $\sqrt{\overline{p'}}$ | 4.39 | 6.42 | 0.68 | 0.49 | -8.19 | 16.98 |
| ln(#statistics) | 0.85 | 0.34 | 2.49 | 0.01 | 0.18 | 1.52 |
| $\hat{\theta}$ | 1.20 | 0.66 |  |  |  |  |
| AIC | 137.32 |  |  |  |  |  |

*Note*:

$\theta$ is the dispersion parameter, with $\text{Var}(Y) = \mu + \frac{\mu^2}{\theta}$, where Y is the number of consistency errors per paper. $\theta$ can vary from 0 to $+\infty$. The smaller $\theta$, the more overdispersion. SPSS reports the inverse value: $k = \frac{1}{\theta}$. In the original paper, Wicherts et al. reported $\hat{k} = 0.83$, which is equal to $\hat{\theta} = \frac{1}{1.20}$.

means that the expected number of consistency errors decreased by 56% when data were shared. The result of the second hypothesis was exactly reproduced in both SPSS and in R ($p = 0.015$, two-tailed, Table 3 represents the contingency table).

Considering all the above-mentioned elements, we classify the original results from Wicherts et al. [6] as strongly reproducible with author assistance: The reproduction results in the correct numbers (i.e., the reproduction results were similar to the original results) and there is low vagueness because of the provided SPSS syntax and R script.

**Multiverse analysis.** For the first hypothesis, the 95% confidence interval of the exponentiated estimate of the regression weight for the target predictor *shared* is depicted in Fig 1. Out of the 32 different specifications considered, 24 (75%) yielded a statistically significant *p*-value (i.e., the 95% confidence interval of the exponentiated estimate did not contain 1). The use of a standard variance estimator increased the standard error estimates, which resulted in larger confidence intervals (depicted by the light grey lines in Fig 1) and thus larger *p*-values. As a result, the statistical significance of the regression weight for the target predictor *shared* depended strongly on the choice of either a robust or a standard variance estimator. Of the 16 analyses using a robust variance estimator, all produced a significant result, whereas only 8 of the 16 analyses using a standard variance estimator produced a significant result. The removal of a single retracted paper (see Section 2 in S1 File) resulted in slightly smaller estimates.

The results of the multiverse analysis for the second hypothesis can be found in Table 4. Each statistical test focusing on the presence of decision errors revealed a statistically significant relationship between data sharing and decision errors. However, when considering the number of decision errors, only the *G*-test revealed a significant result. In sum, the statistical relationship between data sharing and the presence of decision errors seems to hold across different reasonable analytical pathways but diminishes if the number of decision errors per paper is used instead of the presence of decision errors.

**Table 3. Data sharing and presence of decision errors per paper.**

|  | **no decision errors** | **at least one decision error** |
|---|---|---|
| not shared | 21 | 7 |
| shared | 21 | 0 |

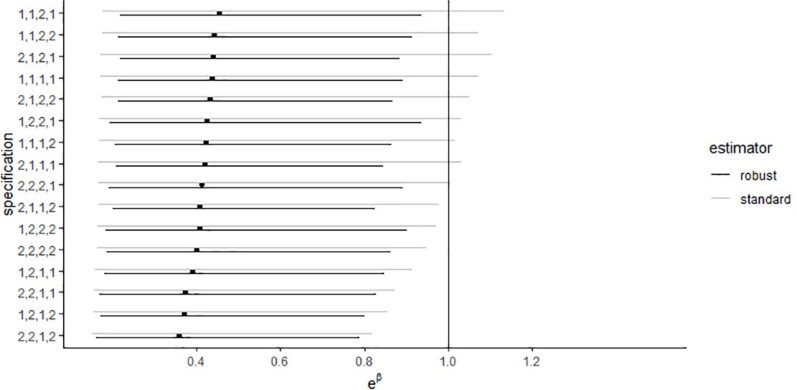

**Fig 1. Multiverse analysis of the regression of consistency errors on data sharing in the original data set from Wicherts et al.** The figure represents the point estimates and 95% confidence intervals for the exponentiated regression coefficients of the predictor *shared* for all combinations of analytical choices. The vertical line represents the absence of an effect ($e^0 = 1$). The choice of variance estimator is indicated in black (robust) or grey (standard). On the y-axis, the other specifications are shown through a four-digit binary code. Each position refers to a decision and the specific number indicates which option has been chosen. The other four decision variables are: (1) natural logarithm of the number of statistics (1 = included in the model as predictor, 2 = included as exposure variable), (2) square root of the mean of the recalculated *p*-values (1 = included in model, 2 = not included), (3) journal (1 = not included in model, 2 = included), and (4) construction of the data set (1 = original data, 2 = original data without retractions). The original analysis is specification 1,1,1,1 in black.

## The replication study with data from Vanpaemel and colleagues [3]

To find out whether data sharing practices had changed since the study of Wicherts et al. [2], Vanpaemel et al. [3] requested research data for a Bayesian reanalysis from empirical papers published in four different APA journals in 2012. It turned out that 38% of authors shared data upon request. The data from Vanpaemel et al. [3] allowed us to conduct a replication of the original study by Wicherts et al. [6]. We again tested the hypotheses that there is a relation between data sharing and consistency errors and that there is a relation between data sharing and decision errors. Additionally, we conducted a multiverse analysis to assess the robustness of our results to alternative analytical decisions. Table 5 provides an overview of the similarities and differences between the original and the current study.

### Materials and methods

**Sample and variables.** Vanpaemel et al. [3] requested data from 394 papers that were published in 2012 in *Emotion*, *Experimental and Clinical Psychopharmacology*, *Journal of*

**Table 4. Results of the multiverse analysis on association between decision errors and data sharing in original data set.**

| data | Chi squared | | Fisher's exact | | G-test | |
|---|---|---|---|---|---|---|
| | **presence** | **count** | **presence** | **count** | **presence** | **count** |
| original data | 0.039 | 0.106 | 0.015 | 0.075 | 0.003 | 0.034 |
| original data w/o retraction | 0.035 | 0.095 | 0.014 | 0.059 | 0.003 | 0.030 |

*Note*:

This table reports *p*-values resulting from 12 contingency tests, testing whether there is a relationship between data sharing on the one hand, and the presence (columns indicated with "presence") or the number of consistency errors (columns indicated with "count") per paper. The first column indicates the construction of the data set (1 = original data, 2 = original data without retractions). The original analysis is a Fisher's exact test on the presence of decision errors and data sharing in the original data (first row, third column).

**Table 5. Similarities and differences between the original study and the current study.**

| aspect | original study | current study |
|---|---|---|
| data | Wicherts, et al. (2006) | Vanpaemel, et al. (2015) |
| year of papers | 2004 | 2012 |
| journals | 2 (out of 4) APA journals | 4 APA journals |
| inclusion criteria | papers from JPSP and JEP:LMC (not further specified) | papers with at least one *p*-value for which there is a Bayesian equivalent that is also detected by statcheck |
| number of papers | 49 | 394 (selection: 286) |
| consistency errors | manual detection of unique NHST results in the main text and in tables of results section and evaluation of test statistics, degrees of freedom and *p*-values | automated detection and evaluation of test statistics, degrees of freedom and *p*-values in APA format in the full paper |
| test statistics | *t*, *F*, $\chi^2$ | *t*, *F*, $\chi^2$ (and *r*, *Z*, *Q* in multiverse analysis) |
| *p*-values | $p < .05$ | $p < .05$ (and $p \leq .05$ as well as all *p*-values in multiverse analysis) |
| number of triplets | 1148 | 4803 (selection: 3175 unique triplets, which varies in the multiverse analysis) |

*Abnormal Psychology*, or *Psychology and Aging* and contained at least one *p*-value for which it is possible to calculate a Bayesian equivalent. Data were shared for 148 (38%) papers.

All papers were subjected to an analysis with *statcheck* [11]. The R package *statcheck* detects triplets of the test statistic, the degrees of freedom, and the *p*-value and evaluates their consistency. This required us to first download all papers from the 2012 issues of the four journals as text files. Next, *statcheck* detected all *t*, *F*, $\chi^2$, *r*, *z*, and *Q* statistics that were reported in the entire paper conform to APA reporting standards, recomputed the *p*-values and flagged consistency and decision errors. Note that *statcheck* also included statistics that (approximately) follow a $\chi^2$ distribution as $\chi^2$ statistics, such as the *H* statistic of the Kruskal-Wallis test. We did not exclude these statistics. We manually reviewed the *statcheck* output for false positives, that is, all flagged consistency errors (and thereby also decision errors) were checked in the papers. False positives could occur due to corrections for multiple testing, one-tailed tests being treated as two-tailed, or misreading (e.g., if authors used a comma instead of a decimal point as the delimiter in a decimal number, see Section 5 in S1 File). Wicherts et al. [6] reported that they only selected unique statistics. Therefore, we excluded duplicate statistics reported within the same paper (Section 6 in S1 File includes an analysis with duplicate statistics). Finally, analogous to Wicherts et al. [6], we selected *t*, *F*, and $\chi^2$ statistics that were reported with a *p*-value below .05, to create the replication data set for the main analyses.

**Statistical analyses.** As with the original data, two sets of statistical analyses on the data from Vanpaemel et al. [3] were conducted. In the first set of analyses, the main replication analyses, we performed, for each of the two hypotheses the same analyses as the ones used for the reproduction of the original results, but this time on the replication data. Replication success was assessed based on the categorization suggested by LeBel et al. [24]: (1) whether the result is statistically significant (i.e., a signal being), (2) whether the 95% confidence interval of the effect size in the replication study includes the original effect size point estimate (i.e., consistency), and (3) whether the replication effect size estimate is more precise.

Second, we performed a multiverse analysis to assess the robustness of the results to alternative analytical decisions, for both hypotheses. We considered the same data analytical and data processing nodes as detailed in the previous section on the original study by Wicherts et al. [6] with the following changes. First, there were no retracted papers in the replication data (see

Section 2 in S1 File), so the removal of retracted papers was not a dimension of variation. Second, we added two possible decisions. The first decision concerned the selection of test statistics during the construction of the data set. Instead of only considering $t$, $F$, and $\chi^2$ statistics, like in the original study, we also included $Q$, $r$, and $z$ statistics. The second decision was related to the selection of $p$-values. Not only did we consider the option of restricting the analysis to $p$-values that were strictly smaller than .05 (i.e., $p < .05$), but we constructed a data set that also included $p$-values equal to .05 (as usually the threshold for statistical significance is not strict), and considered a third option in which we included all $p$-values. In sum, these last two decisions created six possible ways to construct the data set. This means that, for the first hypothesis, together with the 16 previous specifications, our decisions gave rise to a total of 96 specifications. Also for this multiverse analysis, we have initially considered additional pathways including other count model families but removed them due to computational and model fit issues (for transparency, the preliminary results are again reported in Section 4 in S1 File). For the second hypothesis, we constructed a multiverse with $6 \times 6 = 36$ reasonable analyses. Note that decision errors in the original and replication data always concern cases in which the reported result is statistically significant and the recalculated result is not (assuming $\alpha = .05$). In the data sets in the multiverse that include all $p$-values, decision errors can also go in the other direction.

## Results

**Main replication analysis.** We downloaded all 440 papers of the 2012 issues as text files and subjected them to a *statcheck* analysis. Next, we only considered the 394 papers included in the study by Vanpaemel et al. [3] (i.e., those that report at least one $p$-value for which a Bayesian equivalent could be calculated). *Statcheck* did not detect any of the $p$-values reported in 81 papers. In the remaining 313 papers, *statcheck* retrieved 4803 triplets, of which 4757 were unique (see Section 6 in S1 File). Upon correction of 40 false positives, of which 14 were incorrectly flagged as decision errors, we found 337 (7%) consistency errors, of which 44 (1%) were decision errors. *Statcheck* retrieved 3175 unique triplets with $t$, $F$, $\chi^2$ statistics reported with a $p$-value below .05 in 286 papers, meaning that there were no such statistics detected in another 27 papers. Out of these 3175 triplets, 197 (6%) contained consistency errors, of which 25 (1%) were decision errors present in 21 papers. For 108 (38%) out of the 286 papers, the data were shared.

The results of the negative binomial regression are shown in Table 6. The relation between consistency errors and data sharing could not be replicated. According to the criteria suggested by LeBel et al. [24] there is no signal (i.e., no statistical significance), and the replication result is inconsistent and more precise. If the data were shared, the expected number of consistency errors only decreased with 17% (for $e^\beta = 0.83$). We also found no statistically significant

**Table 6. Replication of the relationship between data sharing and consistency errors.**

|  | $\hat{\beta}$ | SE | z | p | 2.5% | 97.5% |
|---|---|---|---|---|---|---|
| intercept | -3.55 | 0.44 | -8.08 | <.001 | -4.41 | -2.69 |
| shared | -0.18 | 0.23 | -0.78 | 0.43 | -0.64 | 0.27 |
| $\sqrt{\bar{p}}$ | 12.62 | 3.06 | 4.12 | <.001 | 6.62 | 18.62 |
| ln(#statistics) | 0.88 | 0.12 | 7.57 | <.001 | 0.65 | 1.11 |
| $\hat{\theta}$ | 0.83 | 0.19 |  |  |  |  |
| AIC | 584.94 |  |  |  |  |  |

**Table 7. Data sharing and presence of decision errors per paper.**

|  | no decision errors | at least one decision error |
|---|---|---|
| not shared | 164 | 14 |
| shared | 101 | 7 |

relationship between data sharing and the presence of decision errors ($p = 0.816$, two-tailed, Table 7 represents the contingency table).

**Multiverse analysis.** Fig 2 shows the results of the multiverse analysis of the first hypothesis. The confidence interval of the exponentiated estimate of the predictor *shared* is depicted for various specifications. Among the 96 specifications, only 7 (7%) resulted in a significant *p*-value. Compared to the multiverse on the original data, the choice for a robust or a standard variance estimator had little impact on the results. For the second hypothesis, the results of the alternative contingency tests can be found in Table 8. None of the 36 tests resulted in a statistically significant relationship between data sharing and the presence or number of decision errors.

## Discussion

In their seminal study, Wicherts et al. [6] found a negative relation between data sharing and both consistency and decision errors. However, Nuijten et al. [9] did not find support for these

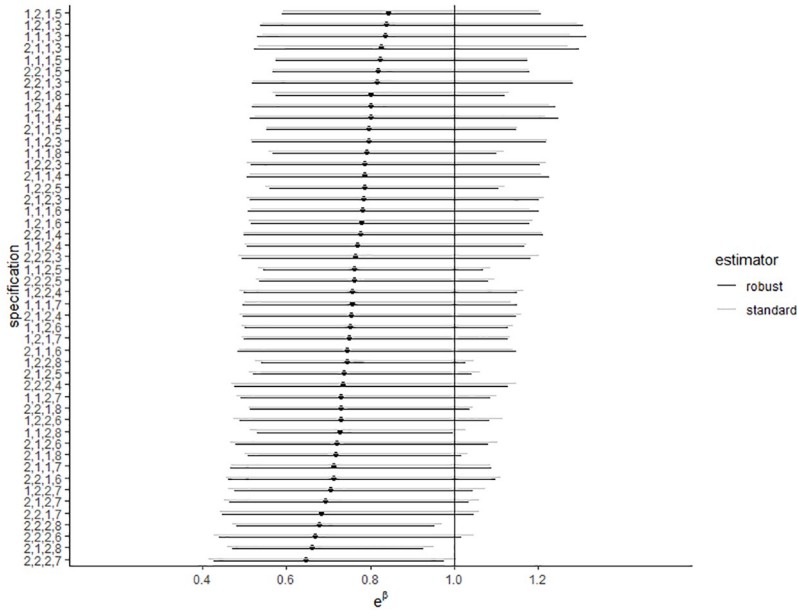

**Fig 2. Multiverse analysis of the regression of consistency errors on data sharing in the replication data set.** The figure represents the point estimates and 95% confidence intervals for the exponentiated regression coefficients of the predictor *shared* for all combinations of analytical choices. The vertical line represents the absence of an effect ($e^0 = 1$). The choice of variance estimator is indicated in black (robust) or grey (standard). On the y-axis, the other specifications are shown through a four-digit code. Each position refers to a decision and the specific number indicates which option has been chosen. The four decision variables are: (1) natural logarithm of the number of statistics (1 = included in the model as predictor, 2 = included as exposure variable), (2) square root of the mean of the recalculated *p*-values (1 = included in model, 2 = not included), (3) journal (1 = not included in model, 2 = included), and (4) construction of the data set (3 = replication data, 4 = the selected *p*-values are less than or equal to .05, 5 = all *p*-values are selected, 6 = all test statistics are selected, 7 = all test statistics and *p*-values that are less or equal to .05 are selected, and 8 = all test statistics and all *p*-values are selected). The main analysis is specification 1,1,1,3 in black.

**Table 8. Multiverse analysis on association between decision errors and data sharing in the replication data set.**

| data | Chi-squared | | Fisher's exact | | G-test | |
|---|---|---|---|---|---|---|
| | presence | count | presence | count | presence | count |
| replication data | 0.84 | 0.61 | 0.82 | 0.84 | 0.66 | 0.41 |
| $p \leq .05$ | 0.48 | 0.46 | 0.39 | 0.64 | 0.34 | 0.26 |
| all $p$-values | 0.23 | 0.50 | 0.18 | 0.54 | 0.15 | 0.42 |
| all stats | 0.79 | 0.50 | 0.82 | 0.71 | 0.62 | 0.29 |
| all stats and $p \leq .05$ | 0.45 | 0.39 | 0.39 | 0.58 | 0.33 | 0.19 |
| all stats and all $p$-values | 0.14 | 0.33 | 0.13 | 0.40 | 0.09 | 0.23 |

*Note*:

This table reports *p*-values resulting from 36 contingency tests, testing whether there is a relationship between data sharing on the one hand, and the presence (columns indicated with "presence") or the number of consistency errors (columns indicated with "count") per paper. The first column indicates the construction of the data set (3 = replication data, 4 = the selected *p*-values are less than or equal to .05, 5 = all *p*-values are selected, 6 = all test statistics are selected, 7 = all test statistics and *p*-values that are less or equal to .05 are selected, and 8 = all test statistics and all *p*-values are selected). The replication analysis is a Fisher's exact test on the presence of decision errors and data sharing in the replication data (first row, third column).

conjectures. In an attempt to shed some light on these discrepancies, we examined the reproducibility, robustness, and replicability of Wicherts et al.'s findings. We focused on these two hypotheses: (1) There is a negative relation between data sharing and the prevalence of consistency errors, and (2) there is a negative relation between data sharing and decision errors. First, we successfully reproduced the original results reported in Wicherts et al. [6], for both hypotheses. Second, a multiverse analysis of the original data revealed that in most specifications the original conclusions hold. However, the results for the first hypothesis were not robust against the choice of a standard variance estimator instead of a robust variance estimator in the negative binomial model. For the second hypothesis, the association between data sharing and decision errors was not robust against working with the number of decision errors instead of the presence of decision errors. Third, we focused on replicability, relying on data from the study by Vanpaemel et al. [3]. Triplets of the test statistic, the degrees of freedom, and the *p*-value were detected and evaluated by *statcheck* (with a manual correction for false positives). Our replication results did not support the original findings: We have found no evidence for a relation between data sharing and (consistency or decision) errors. Fourth, also under several alternative analytical decisions, we could not obtain evidence for such a relation.

In our analyses, we did not exploit researcher degrees of freedom, because we applied the same analyses as Wicherts et al. [6], and even investigated the robustness of the outcomes to alternative analytical choices. Nevertheless, clear preregistered expectations regarding the hypotheses under test would have allowed for stronger theoretical conclusions. For instance, we did not specify whether we expect the relationship between data sharing and consistency errors would change over time or whether the method by which consistency errors are retrieved would matter. In other words, there are no critical elements predefined under which the hypotheses would theoretically hold. This complicates the distinction between direct and conceptual replication [25], even though we tried to stay as close as possible to the original methods.

In the following subsections, we delve into two major differences between the original and the replication study that can explain the differences in results. First, the two samples differed in the papers they included, and are possibly subjected to sampling bias. Second, there were differences between the detection and evaluation of triplets of the test statistic, the degrees of freedom, and the *p*-value.

## Papers included in the samples

There are several systematic differences between the original and replication sample. First of all, the papers in the two samples were published in two different years, 2004 and 2012. Awareness of (the need for) transparent research practices has been growing in the past years, so the replication sample might show a cohort effect. Although the data-sharing rate and the rate of consistency and decision errors were very similar in the two samples, the relationship between data sharing and consistency errors has possibly changed over time. However, also note that the original sample consisted of papers from journals with the highest data-sharing rate, whereas no such selection took place for the replication sample. Second, all papers came from different journals. All journals were APA journals and although the topics of some papers overlapped in both samples (e.g., studies on personality and social psychology), other papers had very different topics. In particular, the original sample contained studies on human cognition, while the replication sample had studies on experimental psychology, clinical psychology, and developmental psychology. Besides topics, the specific journal policies (including guidelines regarding data sharing) probably differed between journals. Also, the data-sharing rate differed between journals [3, 6]. The few models in the multiverse analysis on the replication data that rendered a statistically significant result, controlled for the journal the paper was published in, which could be an indication of differences between journals.

## Detection and evaluation of triplets

Another factor that might have contributed to the differences in results between the original and the replication study is the way triplets were extracted and evaluated from the papers. Perhaps *statcheck* is not a good proxy for the original manual search for consistency errors by Wicherts et al. [6]. The manual method and *statcheck* have been compared before in a validity study by Nuijten et al. [12]. They found that *statcheck* read 67.5% of the triplets that were manually detected by Wicherts et al. [6], but evaluated more triplets as consistency errors. The inter-rater reliability between *statcheck* and the manual assessment for triplets extracted by both methods varied from good (.69) to excellent (.89) [26]. The exact score depended on the specific *statcheck* settings and the inclusion of $p = .000$ in the sample (the latter is a consistency error according to *statcheck*, but not according to Wicherts et al. [6]). Nuijten et al. [12] warned about the interpretation of *statcheck* results, due to the overestimation of consistency errors, but they argued that *statcheck* is suitable to give an overall indication of the prevalence of consistency errors in a large body of literature, and that individual results should be checked by hand. Based on these conclusions, we assumed that *statcheck* (together with a manual check for false positives) sufficiently approximated the manual method by Wicherts et al. to replicate their original study.

**What if Wicherts and colleagues had applied *statcheck*?.** When Wicherts et al. [6] conducted their study, *statcheck* did not yet exist. To investigate the actual implications of employing *statcheck* instead of extracting and checking statistics by hand, we repeated the reproduction analyses on another data set. This data set, retrieved from the validity study by Nuijten et al. [12] (https://osf.io/axfbn/), contained *statcheck* results from the papers originally analyzed by Wicherts et al. [6] with an evaluation of consistency and decision errors, both without and with automatic detection of one-tailed tests. Note that in this data set no manual correction for false positives has been done. Since this data set did not include the recalculated *p*-values, we applied *statcheck* without automatic detection of one-tailed tests (as in our main replication analysis) to compute them. Based on these data, there was no longer a statistically significant relation between data sharing upon request and the number of consistency errors, the parameter estimate was reversed in sign for both *statcheck* results without automatic

**Table 9. Presence decision errors and data sharing.**

| | original data | | statcheck | | statcheck one tailed | |
|---|---|---|---|---|---|---|
| | **0** | **$\geq 1$** | **0** | **$\geq 1$** | **0** | **$\geq 1$** |
| not shared | 21 | 7 | 21 | 5 | 18 | 8 |
| shared | 21 | 0 | 17 | 0 | 15 | 2 |
| *p*-value | 0.02 | two-tailed | 0.14 | two-tailed | 0.27 | two-tailed |

*Note*:

This table displays the contingency table of data sharing and the presence of decision errors and the *p*-value resulting from the Fisher's exact test. The first column shows the original results from the original study, the second column shows the results when consistency errors were detected by the R package *statcheck* without automatic detection of one-tailed tests, the last column shows the results when consistency errors were detected by statcheck with automatic detection of one-tailed tests.

detection of one-tailed tests ($\beta = 0.25$, $z = 0.51$, $p = 0.61$), and *statcheck* results with automatic detection of one-tailed tests ($\beta = 0.45$, $z = 0.83$, $p = 0.4$). Full regression outputs are available in the Section 7 in S1 File. Further, the relation between data sharing and decision errors became attenuated when decision errors were evaluated by *statcheck* instead of manually (see Table 9). The drastic change of the conclusions suggests a larger discrepancy between the manual search for consistency errors by Wicherts et al. [6] and *statcheck* (without a manual correction of false positives) than we would have expected based on the results and conclusions from the validity study by Nuijten et al. [12].

**Why do results with the manual check and *statcheck* differ?.** There were two main discrepancies between the manual check and *statcheck*: which statistics were extracted and which statistics were considered to have consistency errors. Nuijten et al. [12] reported that Wicherts et al. [6] extracted 436 statistics that were not detected by *statcheck* (i.e., because the results were not reported in the standard APA format, or reported in a table, or an extra value such as the mean squared error was reported within the triplet). Vice versa, *statcheck* found 63 statistics that were not extracted by Wicherts et al., due to a variety of reasons (e.g., the statistics were not reported in the results section of the paper). In total, 712 statistics were extracted by both *statcheck* and Wicherts et al. [6]. For these statistics, consistency errors (and also the recalculation of *p*-values) could be evaluated in three ways: by a manual assessment (that takes context into account, such as statistical corrections and one-tailed tests), by *statcheck* without automatic one-tailed test detection, and by *statcheck* with automatic one-tailed test detection.

To get more insight into how the two discrepancies (i.e., in extraction and evaluation) resulted in different conclusions regarding the relationship between data sharing and consistency errors, performed the main analyses three more times on the statistics that were both extracted by the manual check and *statcheck*. When the assessment was manual, the relationship between consistency errors and data sharing was consistent with the original finding, but it was not statistically significant ($\beta = -0.77$, $z = -1.93$, $p = 0.054$). This suggests that the detection of triplets played a role. When the assessment was done by *statcheck*, the relationship between consistency errors and data sharing was reversed, and not statistically significant, for both consistency errors evaluated by *statcheck* without automatic one-tailed test detection ($\beta = 0.41$, $z = 0.78$, $p = 0.43$), and by *statcheck* with automatical one-tailed test detection ($\beta = 0.65$, $z = 1.12$, $p = 0.26$). This suggests that how consistency errors were evaluated also played a role.

To see whether this implies that *statcheck* is not a good method to detect and evaluate triplets, we return to the validity study by Nuijten et al. [12]. For triplets detected by both methods, the *statcheck* results contained 27 discrepancies in consistency errors from the original data of Wicherts et al. [6]: Seven in one paper because it evaluated $p = .000$ as a consistency error (while Wicherts et al. [6] did not), 11 in two papers because of a Huyn-Feldt correction (i.e.,

which results in a false positive in *statcheck*), and nine in four papers because it failed to identify a one-tailed test (which also results in a false positive in *statcheck*). With automatic one-tailed detection, *statcheck* only failed to identify one one-tailed test. This means that the original effect is dependent on the triplets in a handful of papers. Moreover, in the current study, we controlled for false positives by manually checking the flagged statistics. Because of this manual correction, it is unlikely that a difference in the evaluation of errors explains the discrepancy between the original and the replication results. Furthermore, there were no statistics reported as $p = .000$ in our sample. However, there remains the possibility that *statcheck* missed statistics that would have an impact on the results. In any case, *statcheck* allowed us to drastically increase the sample size, without an increased risk for human errors or potential biases.

## Conclusions

Wicherts et al. [2] experienced firsthand that researchers are not keen on sharing data for reanalysis when they requested data from papers published in 2004. About a decade later, Vanpaemel et al. reported a slightly higher data-sharing rate, but there was not much improvement, despite an increase in awareness of the importance to share research data due to the replication crisis. Wicherts et al. [6] speculated that authors might not share their data out of fear to expose errors. Because it is impossible to reanalyze unavailable data, they tested this conjecture by investigating the relation between data sharing upon request and consistency errors in papers. Their study was of great importance, not only because of the relationship they found, but also because of the emphasis it put on the ethical imperative for open data. The current study aimed to shed more light on the relationship between data sharing and consistency errors. The first part consisted of a reproduction of the results from the study by Wicherts et al. [6]. These results were strongly reproducible because the authors maintained good data hygiene and additionally shared the original analysis code (which was not requested by us). In addition, the original results were robust against most alternative analytical decisions. Our replication study in the second part of this paper, however, failed to provide support for the hypotheses put forward by Wicherts et al. [6]. Perhaps the discrepancy in results can be explained by systematic differences between the samples, by a difference in the selection of statistical results, or simply by a sampling error. Overall, in our work, we did not find evidence for a strong, general relationship between sharing research data upon request and consistency (and decision) errors. However, this should not lead to the conclusion that sharing research data does not matter, on the contrary. Without raw data, detectable errors can have many sorts of possible underlying causes in a paper. For a thorough verification of the reported results, raw data is necessary. For a thorough verification of the reported results, raw data is necessary."

## Supporting information

**S1 File. Supplemental material.**
(PDF)

## Acknowledgments

We wish to thank Jelte Wicherts for sharing the original data sets, SPSS syntax and R script from his study with us, and for his suggestions. We also thank Michèle Nuijten for sharing her data sets and R scripts on the Open Science Framework.

## Author Contributions

**Conceptualization:** Wolf Vanpaemel, Tom Heyman.

**Formal analysis:** Aline Claesen.

**Investigation:** Anne-Sofie Maerten.

**Supervision:** Wolf Vanpaemel, Francis Tuerlinckx, Tom Heyman.

**Validation:** Thomas Verliefde.

**Writing – original draft:** Aline Claesen.

**Writing – review & editing:** Wolf Vanpaemel, Anne-Sofie Maerten, Thomas Verliefde, Francis Tuerlinckx, Tom Heyman.

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
