## [Decision Letter · Decision Letter 0]

11 Sep 2022

PONE-D-22-16736Data sharing upon request and statistical consistency errors in Psychology: A replication of Wicherts, Bakker and Molenaar (2011)PLOS ONE

Dear Dr. Claesen,

Thank you for submitting your manuscript to PLOS ONE. After careful consideration, we feel that it has merit but does not fully meet PLOS ONE’s publication criteria as it currently stands. Therefore, we invite you to submit a revised version of the manuscript that addresses the points raised during the review process.

Dear Authors,

Thank you for your submission. 

This is a comprehensive and well reported replication study on an important topic. 

It provides valuable information on the relevance of data sharing practices and policies with respect to the reproducibility of research results. 

The reviewers have suggested several minor revisions. Please address these. 

I also have one suggested revision. Reviewer 2 made a comment about acknowledgement of the different journals included in the replication study as this may be helpful to further provide context for the degree of inference that can be made from the replication results. As part of this acknowledgment you may wish to highlight the distinction between a "retest (direct) reproduction attempt" and an "approximate (conceptual) reproduction attempt". This is a distinction described by: Zwaan, R., Etz, A., Lucas, R., & Donnellan, M. (2018). Making replication mainstream. Behavioral and Brain Sciences, 41, E120. doi:10.1017/S0140525X17001972. It is also summarized well by Niven, D.J., McCormick, T.J., Straus, S.E. et al. Reproducibility of clinical research in critical care: a scoping review. BMC Med 16, 26 (2018). https://doi.org/10.1186/s12916-018-1018-6. 

This differentiation may help to also frame the methodological approach you have taken.

We look forward to receiving your revised manuscript.

Kind regards,

Niklas Bobrovitz

Academic Editor

PLOS ONE

Journal Requirements:

Reviewers' comments:

Reviewer's Responses to Questions

**Comments to the Author**

1. Is the manuscript technically sound, and do the data support the conclusions?

Reviewer #1: Yes

Reviewer #2: Yes

2. Has the statistical analysis been performed appropriately and rigorously? 

Reviewer #1: Yes

Reviewer #2: Yes

3. Have the authors made all data underlying the findings in their manuscript fully available?

Reviewer #1: No

Reviewer #2: Yes

4. Is the manuscript presented in an intelligible fashion and written in standard English?

Reviewer #1: Yes

Reviewer #2: Yes

5. Review Comments to the Author

Reviewer #1: In this submission, the authors report assessments of the reproducibility, robustness, and replicability of the association between data sharing upon request and misreported statistical results as documented by me and my co-authors in 2011. This work is meticulous, reported well, and the conclusions are clearly supported by the data and analyses. This submission is of very high quality and meets all PLOS ONE publication criteria. Just a few minor issues.

1) neither the original study nor the replication were preregistered, while two of studies in Nuijten et al. (2017) were preregistered. Although the multiverse analyses shed some light on the robustness of findings, the use of preregistration would have improved these studies. This could be discussed.

2) I couldn't access the OSF files without requesting access. I agree with the authors that the sharing of data on data sharing behaviours represents an ethical risk and so it is understandable that they could not put all the data in the open.

3) multiverse analyses presume that analytic choices made are mostly arbitrary. This perhaps warrants some discussion.

Signed,

Jelte Wicherts

Reviewer #2: PONE-D-22-16736

Thank you for the opportunity to review this manuscript. Replication studies, especially those regarding issues with data sharing, are both timely and important. The manuscript was well-written and replication was comprehensive while being transparent. Overall, the manuscript would benefit from minimal revisions mainly aiming at contextualizing the impact of the study.

1. One reason Wicherts et al. (2011) conducted their study is because it is an ethical imparative for psychologists to share their data upon request by other psychologists. Noting the importance of why authors' reluctance to share data and syntax/code for verifying results would strengthen the impact of the manuscript. From an ethics standpoint, Bosma & Granger (2022) provide a recent commentary on the ethical implications related to data sharing, which could be helpful for contextualizing why this manuscript is important.

Bosma, C. M., & Granger, A. M. (2022). Sharing is caring: Ethical implications of transparent research in psychology. American Psychologist.

2. Although it is noted in the discussion that Wicherts et al. results did not replicate with the new sample, more discussion can be provided on considering potential cohort effects. Awareness of open data practices have been growing over the past ~15 years and researchers may be more careful about reporting results since Wicherts et al and others began shedding more light on this issue. Though this cannot be tested given the data available to the authors, it is worth discussing.

3. More discussion should be provided regarding sampling bias. Transparency in research varies greatly between sub-disciplines in psychology, yet the the authors attempted replicating Wicherts et al. using convenient data from different journals. Further, although APA journals fall under one organization, the policies for each journal differ greatly in addition to sub-discipline norms. Greater acknowledgement of the different journals included in the replication study would be helpful to further provide context for the degree of inference that can be made from the replication results.

4. Building on comment 1, the authors can add several statements in the introduction and conclusion for why the study broadly matters. Psychologists are ethically supposed to share their data for verification as part of ensuring that what is reported is accurate. The bigger story is that psychologists are still reluctant to share their data upon request, but is helpful to know that perhaps reluctance to sharing is not clearly connected with consistency/decision errors.

6. PLOS authors have the option to publish the peer review history of their article (what does this mean?). If published, this will include your full peer review and any attached files.

Reviewer #1: **Yes: **Jelte Wicherts

Reviewer #2: No

---

## [Author Response · Author response to Decision Letter 0]

16 Feb 2023

Response to reviewers

Dear editor,

Thank you for the opportunity to submit a revised version of our manuscript. With the help of the constructive comments, we improved our manuscript. We provide a detailed response to your and the reviewers’ comments below. 

Academic editor: Niklas Bobrovitz

This is a comprehensive and well reported replication study on an important topic. 

It provides valuable information on the relevance of data sharing practices and policies with respect to the reproducibility of research results. 

Thank you for your encouraging words.

The reviewers have suggested several minor revisions. Please address these. 

I also have one suggested revision. Reviewer 2 made a comment about acknowledgement of the different journals included in the replication study as this may be helpful to further provide context for the degree of inference that can be made from the replication results. As part of this acknowledgment you may wish to highlight the distinction between a "retest (direct) reproduction attempt" and an "approximate (conceptual) reproduction attempt". This is a distinction described by: Zwaan, R., Etz, A., Lucas, R., & Donnellan, M. (2018). Making replication mainstream. Behavioral and Brain Sciences, 41, E120. doi:10.1017/S0140525X17001972. It is also summarized well by Niven, D.J., McCormick, T.J., Straus, S.E. et al. Reproducibility of clinical research in critical care: a scoping review. BMC Med 16, 26 (2018). https://doi.org/10.1186/s12916-018-1018-6. 

This differentiation may help to also frame the methodological approach you have taken.

Thank you for your suggestion. We removed the term “exact replication” from the manuscript. However, whether this study is a direct or conceptual replication, depends on which elements of the study are seen as critical to produce the original result, according to Zwaan et al. (2018). This distinction makes sense, however, we did not have a particular a priori expectation, and even if we did, we cannot claim it since we did not preregister this study. Therefore, we opted for referring to Zwaan et al. (2018) and emphasizing the similarities and differences between the studies. 

We added a paragraph on limitations to the discussion section: 

“In our analyses, we did not exploit researcher degrees of freedom, because we applied the same analyses as Wicherts et al. (2011), and even investigated the robustness of the outcomes to alternative analytical choices. Nevertheless, clear preregistered expectations regarding the hypotheses under test would have allowed for stronger theoretical conclusions. For instance, we did not specify whether we expect the relationship between data sharing and consistency errors would change over time or whether the method by which consistency errors are retrieved would matter. In other words, there are no critical elements predefined under which the hypotheses would theoretically hold. This complicates the distinction between direct and conceptual replication (Zwaan et al., 2017), even though we tried to stay as close as possible to the original methods. 

In the following subsections, we delve into two major differences between the original and the replication study that can explain the differences in results. First, the two samples differed in the papers they included, and are possibly subjected to sampling bias. Second, there were differences between the detection and evaluation of triplets of the test statistic, the degrees of freedom, and the p-value.”

Reviewer 1: Jelte Wicherts

In this submission, the authors report assessments of the reproducibility, robustness, and replicability of the association between data sharing upon request and misreported statistical results as documented by me and my co-authors in 2011. This work is meticulous, reported well, and the conclusions are clearly supported by the data and analyses. This submission is of very high quality and meets all PLOS ONE publication criteria. Just a few minor issues.

We wish to thank Dr. Wicherts for the constructive comments. We address the issues raised below.

1) neither the original study nor the replication were preregistered, while two of studies in Nuijten et al. (2017) were preregistered. Although the multiverse analyses shed some light on the robustness of findings, the use of preregistration would have improved these studies. This could be discussed.

We added a paragraph to the discussion section, which contains the following on preregistration: “In our analyses, we did not exploit researcher degrees of freedom, because we applied the same analyses as Wicherts et al. (2011), and even investigated the robustness of the outcomes to alternative analytical choices. Nevertheless, clear preregistered expectations regarding the hypotheses under test would have allowed for stronger theoretical conclusions.”

2) I couldn't access the OSF files without requesting access. I agree with the authors that the sharing of data on data sharing behaviours represents an ethical risk and so it is understandable that they could not put all the data in the open.

Thank you for checking, something went wrong with including the links. This problem should be fixed now. Note that, in the submission form we stated that we will provide repository information for the data at acceptance. The OSF project is still private and you need a separate link (provided in the manuscript) to access each component.

3) multiverse analyses presume that analytic choices made are mostly arbitrary. This perhaps warrants some discussion.

In the introduction we removed the following: “We then continue with an investigation of the robustness of their results. Because their analysis, like all statistical analyses, featured a number of arbitrary decisions, we carried out several alternative, analytical approaches to evaluate the robustness of the results against other justifiable analytical decisions (i.e., a multiverse analysis; see Steegen et al., 2016).”

We replaced it with (and added a reference): “Because there are often numerous viable ways to conduct a data analysis (i.e., researcher degrees of freedom, Wicherts et al., 2016), any data analysis which reports a single pathway provides an incomplete picture. We continue with an investigation of the robustness of their results against other reasonable analytical decisions, by carrying out several alternative analytical approaches (i.e., a multiverse analysis; see Steegen et al., 2016).”

Reviewer 2

Thank you for the opportunity to review this manuscript. Replication studies, especially those regarding issues with data sharing, are both timely and important. The manuscript was well-written and replication was comprehensive while being transparent. Overall, the manuscript would benefit from minimal revisions mainly aiming at contextualizing the impact of the study.

We wish to thank Reviewer 2 for the constructive comments. We address their suggestions below.

1. One reason Wicherts et al. (2011) conducted their study is because it is an ethical imparative for psychologists to share their data upon request by other psychologists. Noting the importance of why authors' reluctance to share data and syntax/code for verifying results would strengthen the impact of the manuscript. From an ethics standpoint, Bosma & Granger (2022) provide a recent commentary on the ethical implications related to data sharing, which could be helpful for contextualizing why this manuscript is important.

Bosma, C. M., & Granger, A. M. (2022). Sharing is caring: Ethical implications of transparent research in psychology. American Psychologist.

Thank you for the suggested reference. We included it in the beginning of the introduction: “A major benefit of sharing research data is that it allows the scientific community to verify the empirical results and scientific findings, and to further build upon the published work. Moreover, as Bosma and Granger (2022) showed, there are also ethical implications of data sharing. Despite all these benefits, Wicherts et al. (2006) found that researchers are not keen on sharing their data when they requested data for reanalysis to investigate the influence of outliers, as they received data from only 27% of the contacted authors.”

We further emphasized the ethical implication in the discussion. We added a sentence regarding the original study by Wicherts et al. (2011): “Their study was of great importance, not only because of the relationship they found, but also because of the emphasis it put on the ethical imperative for open data.”

2. Although it is noted in the discussion that Wicherts et al. results did not replicate with the new sample, more discussion can be provided on considering potential cohort effects. Awareness of open data practices have been growing over the past ~15 years and researchers may be more careful about reporting results since Wicherts et al and others began shedding more light on this issue. Though this cannot be tested given the data available to the authors, it is worth discussing.

We could have indeed discussed this a bit more. We adapted the discussion part (see our reply to the next comment), and added: ”Awareness of (the need for) transparent research practices has been growing in the past years, so the replication sample might show a cohort effect.“

3. More discussion should be provided regarding sampling bias. Transparency in research varies greatly between sub-disciplines in psychology, yet the the authors attempted replicating Wicherts et al. using convenient data from different journals. Further, although APA journals fall under one organization, the policies for each journal differ greatly in addition to sub-discipline norms. Greater acknowledgement of the different journals included in the replication study would be helpful to further provide context for the degree of inference that can be made from the replication results.

This is a good point, although we do not believe that the replication data are the result of convenience sampling. Vanpaemel et al. (2015) contacted authors in the same way as Wicherts et al. (2015) in similar APA journals. 

In the discussion section, we added: “In the following subsections, we delve into two major differences between the original and the replication study that can explain the differences in results. First, the two samples differed in the papers they included, and are possibly subjected to sampling bias. Second, there were differences between the detection and evaluation of triplets of the test statistic, the degrees of freedom, and the p-value.”

In our manuscript, we already specified the differences in topics. Based on this comment, we also acknowledge that the specific journal policies could have differed at the time. We added the following to the subsection about the papers included in the samples: “Besides topics, the specific journal policies (including guidelines regarding data sharing) probably differed between journals.”

4. Building on comment 1, the authors can add several statements in the introduction and conclusion for why the study broadly matters. Psychologists are ethically supposed to share their data for verification as part of ensuring that what is reported is accurate. The bigger story is that psychologists are still reluctant to share their data upon request, but is helpful to know that perhaps reluctance to sharing is not clearly connected with consistency/decision errors.

We added the following to the conclusions: “Overall, in our work, we did not find evidence for a strong, general relationship between sharing research data upon request and consistency (and decision) errors. However, this should not lead to the conclusion that sharing research data does not matter, on the contrary. Without raw data, detectable errors can have many sorts of possible underlying causes in a paper. For a thorough verification of the reported results, raw data is necessary. For a thorough verification of the reported results, raw data is necessary.”

---

## [Editor Report · Decision Letter 1]

27 Mar 2023

Data sharing upon request and statistical consistency errors in Psychology: A replication of Wicherts, Bakker and Molenaar (2011)

PONE-D-22-16736R1

Dear Dr. Claesen,

We’re pleased to inform you that your manuscript has been judged scientifically suitable for publication and will be formally accepted for publication once it meets all outstanding technical requirements.

Kind regards,

Niklas Bobrovitz

Academic Editor

PLOS ONE
---

## [Editor Report · Acceptance letter]

30 Mar 2023

PONE-D-22-16736R1 

Data sharing upon request and statistical consistency errors in Psychology: A replication of Wicherts, Bakker and Molenaar (2011) 

Dear Dr. Claesen:

I'm pleased to inform you that your manuscript has been deemed suitable for publication in PLOS ONE. Congratulations! Your manuscript is now with our production department. 

Kind regards, 

on behalf of

Dr. Niklas Bobrovitz 

Academic Editor

PLOS ONE